# "Are Your Explanations Reliable?" Investigating the Stability of LIME in Explaining Text Classifiers by Marrying XAI and Adversarial Attack

**Christopher Burger**
University of Mississippi
Oxford, MS, USA
cburger@olemiss.edu

**Lingwei Chen**
Wright State University
Dayton, OH, USA
lingwei.chen@wright.edu

**Thai Le**
University of Mississippi
Oxford, MS, USA
thaile@olemiss.edu

## Abstract

LIME has emerged as one of the most commonly referenced tools in explainable AI (XAI) frameworks that is integrated into critical machine learning applications–e.g., healthcare and finance. However, its stability remains little explored, especially in the context of text data, due to the unique text-space constraints. To address these challenges, in this paper, we first evaluate the inherent instability of LIME on text data to establish a baseline, and then propose a novel algorithm XAIFOOLER to perturb text inputs and manipulate explanations that casts investigation on the stability of LIME as a text perturbation optimization problem. XAIFOOLER conforms to the constraints to preserve text semantics and original prediction with small perturbations, and introduces Rank-biased Overlap (RBO) as a key part to guide the optimization of XAIFOOLER that satisfies all the requirements for explanation similarity measure. Extensive experiments on real-world text datasets demonstrate that XAIFOOLER significantly outperforms all baselines by large margins in its ability to manipulate LIME's explanations with high semantic preservability. The code is available at https://github.com/cburgerOlemiss/XAIFooler

## 1 Introduction

Machine learning has witnessed extensive research, leading to its enhanced capability in predicting a wide variety of phenomena (Pouyanfar et al., 2018). Unfortunately, this increased effectiveness has come at the cost of comprehending the inner workings of the resulting models. To address this challenge, explainable AI (XAI), also known as interpretable AI (Tjoa and Guan, 2020), has emerged as a discipline focused on understanding *why* a model makes the predictions it does. XAI continues to grow in importance due to both legal and societal demands for elucidating the factors contributing to specific model predictions.

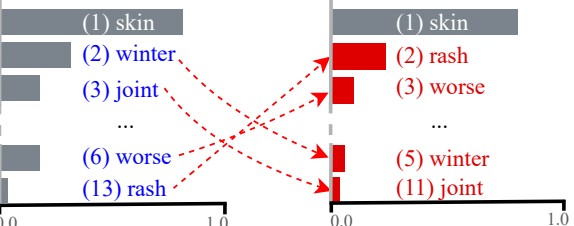

*"I have a skin rash that ~~gets~~ becomes worse in the winter. I have to moisturize more ~~regularly~~ consistently and have [...]. I also have joint pain."*

Figure 1: With only two perturbations, XAIFOOLER significantly demoted "joint pain" symptom from top-3 to 11th-rank (left→right figure) while *maintaining the original prediction label* of "peptic ulcers" and *retaining the clarity and meaning* of the original text.

While explainability in AI as a general concept has yet to fully coalesce to an accepted set of definitions (Ghorbani et al., 2019), the concept of *Stability* starts to occur throughout the discussions and remains prominent (Molnar, 2022; Doshi-Velez and Kim, 2017; Zhang et al., 2020). A stable explanation refers to one where small changes to the input should have a corresponding small effect on the output. In other words, similar inputs to a model should produce similar explanations. Lack of stability in an explanatory method undermines its trustworthiness (Ghorbani et al., 2019), and renders all subsequent explanations suspect. This lack of trustworthiness is one reason why the adoption of AI in disciplines like healthcare has progressed slower than in other disciplines (Markus et al., 2021).

Previous work on XAI stability has focused on models where function-based continuity criteria naturally apply to tabular and image-based data (Alvarez-Melis and Jaakkola, 2018; Zhang et al., 2020; Ghorbani et al., 2019). However, text data in natural language is not so amenable to such direct quantification. Therefore, the process of generating appropriate perturbations to test stability in text explanations remains little explored (Ivankay et al., 2022). Unlike perturbations in tabular or image-based data, text perturbations pose unique chal-

lenges to satisfy necessary constraints of semantic similarity and syntactic consistency. Violation of these constraints can create a fundamentally different meaning conveyed by the document. Under the assumption the explanatory algorithm works correctly, these perturbations may alter the resulting explanation, where quantifying difference between explanations also needs to characterize their properties, such as feature ordering and weights within explanations presented in Fig. 1.

In this paper, we explore the stability of explanations generated on text data via LIME (Ribeiro et al., 2016), which is a widely used explanatory algorithm in XAI frameworks. We first examine the inherent stability of LIME by altering the number of samples generated to train the surrogate model. Using this baseline, we then propose XAIFOOLER, a novel algorithm that perturbs text inputs and manipulates explanations to investigate in depth the stability of LIME in explaining text classifiers. Given a document, XAIFOOLER proceeds with iterative word replacements conforming to the specified constraints, which are guided by Rank-biased Overlap (RBO) that satisfies all the desired characteristics for explanation similarity measure. As such, XAIFOOLER yields advantages that only small perturbations on the least important words are needed, while the semantics and original prediction of the input are preserved yet top-ranked explanation features are significantly shifted. Fig. 1 shows an example that XAIFOOLER only performs two perturbations (i.e., "gets" → "becomes" and "regularly" → "consistently"), but effectively demotes the top-3 features–e.g., "joint" (3rd-rank)→(11th-rank), that explains the "joint pain" symptom without changing the prediction of *peptic ulcers* disease. Our contributions are summarized as follows.

- We assess the inherent instability of LIME as a preliminary step towards better understanding of its practical implications, which also serves as a baseline for subsequent stability analysis.
- We cast investigation on the stability of LIME as a text perturbation optimization problem, and introduce XAIFOOLER with thoughtful constraints and explanation similarity measure RBO to generate text perturbations that effectively manipulates explanations while maintaining the class prediction in a cost-efficient manner.
- We conduct extensive experiments on real-world text datasets, which validate that XAIFOOLER significantly outperforms all baselines by large

margins in its ability to manipulate LIME's explanations with high semantic preservability.

## 2 Background

### 2.1 LIME

In this paper, we adopt Local Interpretable Model-agnostic Explanations (LIME) (Ribeiro et al., 2016) as our target explanatory algorithm. LIME has been a commonly researched and referenced tool in XAI frameworks, which is integrated into critical ML applications such as finance (Gramegna and Giudici, 2021) and healthcare (Kumarakulasinghe et al., 2020; Fuhrman et al., 2022). To explain a prediction, LIME trains a shallow, explainable surrogate model such as Logistic Regression on training examples that are synthesized *within the vicinity* of an individual prediction. The resulting explanation is a subset of *coefficients* of this surrogate model that satisfies the fundamental requirement for interpretability. In NLP, explanations generated by LIME are features–e.g., words, returned from the original document, which can be easily understood even by non-specialists.

### 2.2 XAI Stability

Existing research work on XAI stability has a predominant emphasis on evaluating models using tabular and/or image data across various interpretation methods, which often use small perturbations to the input data to generate appreciable different explanations (Alvarez-Melis and Jaakkola, 2018; Ghorbani et al., 2019; Alvarez-Melis and Jaakkola, 2018), or generate explanations that consist of arbitrary features (Slack et al., 2020).

LIME specifically has been analyzed for its efficacy. Garreau et al. first investigated the stability of LIME for tabular data (Garreau and Luxburg, 2020; Garreau and von Luxburg, 2020), which showed that important features can be omitted from the resulting explanations by changing parameters. They extended the analysis to text data later (Mardaoui and Garreau, 2021) but only on the fidelity instead of *stability* of surrogate models. Other relevant works in text domain include (Ivankay et al., 2022), which utilized gradient-based explanation methods, assuming white-box access to the target model and hence not realistic because model parameters are often inaccessible in practice; and (Sinha et al., 2021), which revealed that LIME's explanations are unstable to black-box text perturbations. However, it adopts a very small sampling rate for LIME

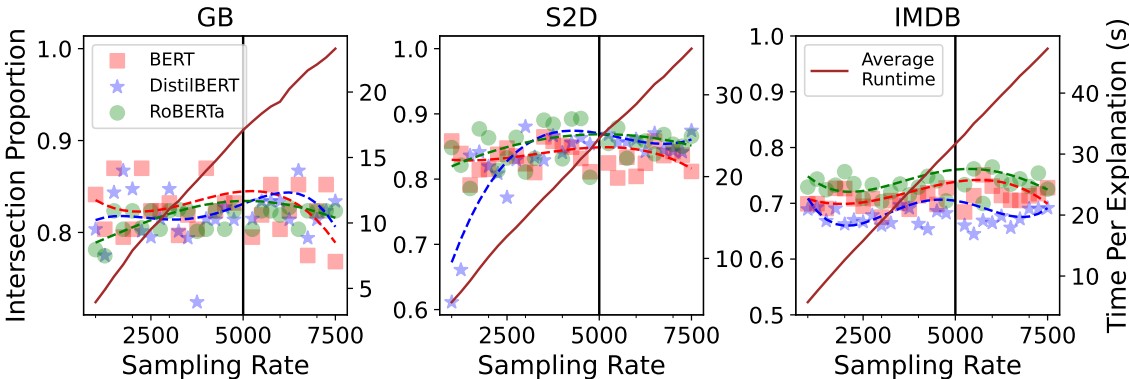

Figure 2: Inherent explanation instabilities for each model and dataset.

($n$=500) which we later demonstrate to significantly overestimate LIME's instability. Moreover, its experiment settings are not ideal as it allows the perturbations of top-ranked predictive features, which naturally change the resulting explanations.

Although existing works have showed that LIME is sensitive to small perturbations in the target model's inputs or parameters, they do not answer the fundamental question of whether LIME itself is inherently unstable *without* any changes to the input or model's parameters. Answers to this *"what is the inherent instability of* LIME*"* question would establish a meaningful baseline that helps us better evaluate our stability analysis.

## 3 Preliminary Analysis

**Inherent Instability of LIME**. This section first assesses the inherent instability of LIME. Our aim is to determine if LIME produces inconsistent results even when no changes are made to the target model's inputs or parameters. Let $d$ be a document whose prediction under a target model $f(\cdot)$ is to be explained. A simplified process of LIME's explanation generation for $f(d)$ is as follows.

- *Step 1:* Generate perturbed document $d_i$ by randomly selecting $k$ words from $d$ and remove all of their occurrences from $d$.
- *Step 2:* Repeat *Step 1* to sample $n$ different perturbations $\{d_i\}_{i=1}^n$.
- *Step 3:* Train an explainable model $g(\cdot)$ via supervised learning with features $d_i$ and labels $f(d_i)$.

Here $n$ is the *sampling rate* which determines the number of local training examples needed to train the surrogate model $g(\cdot)$. Sampling rate $n$ greatly influences the performance of LIME. Intuitively, a small $n$ often leads to insufficient amount of data for training a good $g(\cdot)$. In contrast, a large $n$ may result in better $g(\cdot)$ but also adversely increase the

| Dataset | Mean | Median | Min |
|---------|------|--------|-----|
| GB | 82.0% ($\downarrow \Delta18\%$) | 82.0% | 75.5% |
| S2D | 84.0% ($\downarrow \Delta16\%$) | 84.4% | 72.9% |
| IMDB | 71.5% ($\downarrow \Delta28.5\%$) | 70.4% | 67.3% |

Table 1: Statistics of explanation similarities when using different alternate sampling rates compared against the base explanation with default sampling rate $n$=5$K$.

runtime due to a large number of inference passes required to collect all $f(d_i)$.

To test the inherent instability of LIME, we first select an arbitrary number of documents and generate explanations at different sampling rates $n$ from 1K–7K for three state-of-the-art classifiers, including BERT (Devlin et al., 2019), RoBERTa (Liu et al., 2019), DistilBERT (Sanh et al., 2019), trained on three corpus varying in lengths. Then, we compare them against the base explanation generated with the default sampling rate ($n$=5$K$). Table 1 shows that altering the sampling rate from its default value results in significant dissimilarities in the explanations ($\downarrow$16%–$\downarrow$28.5% in similarity on average), which observe to be more significant on longer documents. This happens because longer documents generally require more training examples to refine the local model $g(\cdot)$ to better approximate $f(\cdot)$. Fig. 2 gives us a closer look into such instability, which further shows evidence that in all datasets, both bounded variation and diminishing returns converge as the sampling rate increases. However, with small sampling rates $n$–e.g., $n$<4K in IMDB dataset, slight changes in $n$ produce significantly dissimilar explanations.

Overall, LIME itself is inherently unstable, especially when $n$ is small, even without any modifications to the target model or its inputs. Contrast to small sampling rates–e.g., $n$=500, adopted in existing analytical works, our analysis implies that $n$ should be sufficiently large according to each

dataset to maximize the trade-off between computational cost and explanation stability. However, the source of the instability within LIME is yet undetermined. We emphasize the importance of this question but leave its investigation for future work. **Motivation** This inherent instability is significant in practice due to the fact that many important AI systems–e.g., ML model debugging (Lertvittayakumjorn and Toni, 2021) or human-in-the-loop system (Nguyen and Choo, 2021), use surrogate methods such as LIME as a core component. This instability also has implications on software security where malicious actors can exploit this instability and force AI models to produce unfaithful explanations. Given that there exists some level of instability within the explanations by default, we now ask if alterations to the input text can demonstrate instability beyond that of the baseline levels already seen. That is, can perturbations to the input that are carefully chosen to retain the underlying meaning of the original text result in appreciably different explanations. If so, the concerns about instability mentioned above become that much more pressing. To do this, we propose to investigate the robustness of LIME by formulating this perturbation process as an adversarial text attack optimization problem below.

## 4 Problem Formulation

Our goal is to determine how much a malicious actor can manipulate explanations generated by LIME via text perturbations–i.e., minimally perturbing a document such that its explanation significantly changes. Specifically, a perturbed document $d_p$ is generated from a base document $d_b$, such that for their respective explanations $e_{d_p}$ and $e_{d_b}$ we *minimize their explanation similarity*:

$$d_p = \underset{d_p}{\operatorname{argmin}} \ \textbf{\textit{Sim}}_\textbf{e}(e_{d_b}, e_{d_p}), \qquad (1)$$

where $\textbf{\textit{Sim}}_\textbf{e}(\cdot)$ is the similarity function between two explanations. To optimize Eq. (1), our method involves a series of successive perturbations within the original document as often proposed in adversarial text literature. In a typical adversarial text setting, malicious actors aim to manipulate the target model's prediction on the perturbed document, which naturally leads to significant changes in the original explanation. But this is not meaningful for our attack; thus, we want to *preserve the original prediction* while altering only its explanation:

$$f(d_b) = f(d_p), \qquad (2)$$

Trivially, changing words chosen arbitrarily and with equally arbitrary substitutions would, eventually, produce an explanation different from the original. However, this will also likely produce a perturbed document $d_p$ whose semantic meaning drastically differs from $d_b$. Thus, we impose a constraint on the semantic similarity between $d_b$ and $d_p$ to ensure that the perturbed document $d_p$ does not alter the fundamental meaning of $d_b$:

$$\textbf{\textit{Sim}}_\textbf{s}(d_b, d_p) \geq \delta, \qquad (3)$$

where $\textbf{\textit{Sim}}_\textbf{s}(\cdot)$ is the semantic similarity between two documents and $\delta$ is a sufficiently large hyperparameter threshold.

Even with the semantic constraint in Eq. (3), there can still be some shift regarding the context and semantics of $d_b$ and $d_p$ that might not be impeccable to humans yet cannot algorithmically captured by the computer either. Ideally, the malicious actor wants the perturbed document $d_p$ to closely resemble its original $d_b$. However, as the number of perturbations grows larger, the document retains less of its original context and meaning. To address this issue, we impose a maximum number of perturbations allowed through the use of a hyperparameter $\epsilon$ as follows:

$$i \leq \epsilon * |f|, \qquad (4)$$

where an accepted $i$-th perturbation will have replaced $i$ total number of words (as each perturbation replaces a single word) and $|f|$ is the total number of *unigram bag-of-words* features in $f$.

We can now generate perturbations in a way that maintains the intrinsic meaning of the original document. If we are to manipulate the explanation (Eq. (1)) while maintaining both Eq. (2), Eq. (3) and Eq. (4), it is trivial to just replace some of the most important features of $f$. However, in practice, changing the most important features will likely result in a violation to constraint in Eq. (2). Moreover, this will not provide meaningful insight to the analysis on stability in that we want to measure how many changes in the perturbed explanation that correspond to small (and not large) alterations to the document. Without the following constraint, highly weighted features will often be perturbed, even with a search strategy focused towards features of minimal impact on the base explanation. This happens due to the strength of the previous constraints focused on maintaining the quality of the document. Thus, the set of top $k$ feature(s) belonging to the base explanation $e_{d_b}$ must appear in

the perturbed explanation $e_{d_p}$:

$$e_{d_p} \cap c \neq \varnothing \quad \forall c \in e_{d_b}[:k]. \qquad (5)$$

Overall, our objective function is as follows.

---

**OBJECTIVE FUNCTION**: Given a document $d_b$, a target model $f$ and hyper-parameter $\delta$, $\epsilon$, $k$, our goal is to find a perturbed document $d_p$ by optimizing the objective function:

$$d_p = \underset{d_p}{\operatorname{argmin}} \ \boldsymbol{Sim}_{\mathbf{e}}(e_{d_b}, e_{d_p}),$$

$$\text{s.t.} \ \ f(d_b) = f(d_p),$$
$$\boldsymbol{Sim}_{\mathbf{s}}(d_b, d_p) \geq \delta, \qquad (6)$$
$$i \leq \epsilon * |f|,$$
$$e_{d_p} \cap c \neq \varnothing \quad \forall c \in e_{d_b}[:k]$$

---

## 5 XAIFOOLER

To solve the aforementioned objective function (Eq. (6)), we propose XAIFOOLER, a novel greedy-based algorithm to manipulate the explanations of a text-based XAI method whose explanations are in the form of a list of words ordered by importance to the surrogate model. We then demonstrate that XAIFOOLER can effectively alter the explanations of LIME beyond the intrinsic instability (Sec. 3) via carefully selected text perturbations.

### 5.1 XAIFOOLER Algorithm

Algorithm 1 describes XAIFOOLER in two steps. First, given an input document, it decides which words to perturb and in what order (Ln. 4). Next, it greedily replaces each of the selected word with a candidate that (i) best minimizes the explanation similarity via $\boldsymbol{Sim}_{\mathbf{e}}$ and (ii) satisfies all the constraints in Sec. 4 until reaching the maximum number of perturbations (Eq. (4)) (Ln. 5–13).

**Step 1: Greedy Word Selection and Ordering.** We disregard all stopwords and the top-$k$ important features based on their absolute feature importance scores returned from LIME. We then order the remaining words in the original document $d_b$ in *descending order* according to how many changes they make in the original prediction when they are individually removed from $d_b$. Intuitively, we prioritize altering features of lower predictive importance to signify the instability of LIME (Eq. 5).

**Step 2: Greedy Search with Constraints.** We subsequently replace each word in the list returned from Step 1 with a list of candidates and only keep those that help decrease the explanation similarity $\boldsymbol{Sim}_{\mathbf{e}}$ (Alg. 1, Ln. 6–10). To satisfy Eq. ( 2, 3,

---

**Algorithm 1** Adversarial Explanation Generation

---

1: *Input:* **target model** $f$, **Original Document** $d_o$, **Base Explanation** $e_b$, **Maximum Perturbation Threshold** $p_t$, **Current Perturbations** $p_c$ **Current Similarity** $s$
2: *Output:* **Perturbed Document** $d_p$, **Perturbed Explanation** $e_p$, **Updated Similarity** $s$
3: *Initialize:* $e_p \leftarrow e_b$, $i \leftarrow 0$, $s \leftarrow 1$, $p_c \leftarrow 0$
4: *Initialize:* I $\leftarrow$ indices of replacement candidates that satisfy $C$
5: **while** $I \neq \varnothing$ *and* $p_c < p_t$ **do**
6:     $s_i = \text{RBO}(d_p, \text{perturb}(d_p[I[i]])$
7:     **if** $s_i < s$ **then**
8:         $s \leftarrow s_i$
9:         $e_p \leftarrow \text{perturb}(d_p[I[i]])$
10:     **end if**
11:     $I[i] = \varnothing$
12:     $i \leftarrow i + 1$
13: **end while**
14: **return** $(d_p, e_p, s)$

---

4, 5), we only accept a perturbation if it satisfies these constraints, and at the same time improves the current best explanation *dissimilarity*. To reduce the search space of replacements, we only select replacement candidates that maintain the same part-of-speech function and also within a similarity threshold with the word to be replaced in counter-fitted Paragram-SL999 word-embedding space (Wieting et al., 2015) as similarly done in prior adversarial text literature (TextFooler (Jin et al., 2020)). Since our perturbation strategy solely applies the constraints at the word level, certain perturbations may result in undesirable changes in textual quality, which is often observed in texts perturbed by popular word-level methods such as TextFooler. However, our proposed algorithm (Alg. 5.1) is generic to any black-box text perturbation functions *perturb*($\cdot$) (Alg. 5.1, Ln. 6, 9).

### 5.2 Explanation Similarity Characterization

The most important factor that guides the optimization of XAIFOOLER is the explanation similarity function $\boldsymbol{Sim}_{\mathbf{e}}$ needed to compare two explanations $e_{d_b}$ and $e_{d_p}$. In selecting a good $\boldsymbol{Sim}_{\mathbf{e}}$ function, we need to first characterize it. The explanations $e_{d_b}$, $e_{d_p}$ returned from LIME are ranked lists. That is, they are an ordered collection of features. Specifically LIME's explanations consist of unique features which are ordered w.r.t their importance to the local surrogate model. While there is a substantial amount of prior work discussing comparison methods for ranked lists, the structure of LIME's explanations and our constraints are not ideal for many common methods. Thus, for our purposes, a desirable $\boldsymbol{Sim}_{\mathbf{e}}$ has properties as follows.

**(A) Positional Importance** (Kumar and Vassilvit-skii, 2010). We require that a relative ordering be imposed on the features–i.e., higher-ranked features should be accorded more influence within $Sim_e(\cdot)$. That is, moving the 1st-ranked feature to rank 10 should be considered more significant than moving the 10th-ranked feature to rank 20. Moreover, this requirement also accounts the fact that end-users often consider only the top $k$ most important *and not all of the* features (Stepin et al., 2021) in practice, and thus $Sim_e(\cdot)$ should be able to distinguish the ranking of different features.

**(B) Feature Weighting.** This is a strengthening of the positional importance by considering not only discrete positional rankings such as 1st, 2nd, 3rd, but also their continuous weights such as the feature importance scores provided by LIME. This provides us more granularity in the resulting explanation similarities and also better signals to guide our optimization process for Eq. (6).

**(C) Disjoint Features.** We require that disjoint lists can be compared. Consider a single word replacement of word $w$ with word $w'$ in a document $d_b$ with base explanation $e_{d_b}$ and perturbed explanation $e_{d_p}$ generated from $d-w+w'$. The perturbed explanation $e_{d_p}$ cannot contain the word $w$ or any subset of $d_b$ containing $w$ as a possible feature. And so each perturbation is likely to result in an explanation that is disjoint. Many similarity measures that have a requirement for non-disjoint lists often have an implementation that relaxes this requirement. However, as the amount of disjoint features increases, these measures will struggle to deliver appropriate similarity values.

**(D) Unequal List Lengths.** Similarly to (C) but if $w'$ already exists in the original explanation $e_b$, the length of $e_p$ will be different from $e_b$ due to $d_p$ having one less unique unigram feature.

### 5.3 Existing Similarity Measures

With the above characterization, we proceed to search for an ideal explanation similarity measure for $Sim_e$ from some of commonly used measures on ranked lists. The first group of such is **weight-based.** This includes **(1) $L_p$:** The standard $L_p$ distances discard the feature order entirely and instead concentrate on their associated weights. Certain formulations exist that allow weighting the elements, but the $L_p$ distances still lack the capacity to handle differing ordered-list lengths. **(2) Center of Mass (COM):** COM is an improvement upon $L_p$

| Feature / Measure | Positional Importance | Feature Weighting | Disjoint Features | Unequal-Length List |
|---|---|---|---|---|
| Jaccard Index | | ✓* | ✓ | ✓ |
| Kendall's $\tau$ | ✓ | ✓* | | |
| Spearman's $\rho$ | ✓ | ✓* | | |
| $L_p$ | | ✓ | ✓ | |
| Center of Mass | | ✓ | ✓ | |
| †RBO | ✓ | ✓ | ✓ | ✓ |

†RBO: Rank-biased Overlap; *: customized formulas exist

Table 2: Comparisons among explanation similarity measures with RBO satisfying all the requirements.

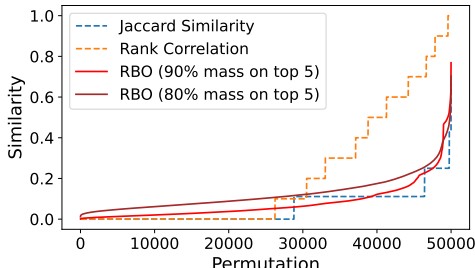

Figure 3: Simulation on 50K permutations of 50 features: RBO outputs smoothed, richer signals while other measures return zero similarity for over half the permutations and show poorer delineation among top-$k$ features, resulting in the non-smoothed, step-wise behavior.

as it not only indicates a change in the weights but also provides information about where that weight is being assigned. However, a change in COM does not imply an actual change in the relative order of features. The second group is *feature-based.* This includes **(1) Jaccard Index**, which compares two sets by computing the ratio of the shared elements between the sets to the union of both sets. Being a strictly set-based measure, the Jaccard index lacks positional importance and feature weighting (extensions exist to allow weight), making it unsuitable for our problem. **(2) Kendall's $\tau$ & Spearman's $\rho$** are commonly used for determining the similarity of two ranked lists. The central idea of both is the order of the features. However, they disregard any ranking weights, the capacity to handle unequal list lengths, and disjoint features. Remedies are also available–e.g., Kumar and Vassilvitskii (2010), but they do not fulfill all the requirements and can result in information loss.

### 5.4 RBO: The Best of both Worlds

So far, none of the mentioned measures satisfies all requirements. Fortunately, there exists *Rank-biased Overlap* (RBO) (Webber et al., 2010) that well fits to our criteria. RBO is a feature-based comparison measure that combines the benefits of

| Dataset/Method | DistilBERT | | | | | BERT | | | | | RoBERTa | | | | |
|---|---|---|---|---|---|---|---|---|---|---|---|---|---|---|---|
| | ABS↑ | RC↑ | INS↓ | SIM↑ | PPL↓ | ABS↑ | RC↑ | INS↓ | SIM↑ | PPL↓ | ABS↑ | RC↑ | INS↓ | SIM↑ | PPL↓ |
| **IMDB** Inherency | 4.92 | 0.34 | 0.83 | 1.00 | 33.17 | 5.66 | 0.48 | 0.83 | 1.00 | 34.51 | 5.90 | 0.52 | 0.82 | 1.00 | 33.88 |
| Random | 5.90 | 0.50 | 0.84 | 0.89 | 71.07 | 6.80 | 0.58 | 0.80 | 0.88 | 75.74 | 7.33 | 0.52 | 0.78 | 0.89 | 73.89 |
| LOM | 3.94 | 0.33 | 0.89 | **0.96** | 40.35 | 4.92 | 0.41 | 0.87 | **0.95** | 43.85 | 6.08 | 0.47 | 0.83 | **0.95** | 43.41 |
| $L_p$ | 9.07 | 0.52 | 0.81 | 0.89 | 69.54 | 8.76 | 0.57 | 0.79 | 0.89 | 71.16 | 9.96 | 0.65 | 0.78 | 0.89 | 72.87 |
| **XAIFOOLER** | **10.43** | **0.65** | **0.75** | 0.89 | 67.01 | **11.44** | **0.69** | **0.72** | 0.89 | 67.79 | **12.65** | **0.76** | **0.72** | 0.90 | 63.15 |
| **S2D** Inherency | 1.30 | 0.23 | 0.88 | 1.0 | 12.3 | 1.65 | 0.24 | 0.85 | 1.0 | 12.30 | 3.09 | 0.36 | 0.76 | 1.00 | 12.30 |
| Random | 1.72 | 0.27 | 0.85 | 0.84 | 77.69 | 2.49 | 0.41 | 0.80 | 0.85 | 79.22 | 3.66 | 0.48 | 0.77 | 0.84 | 83.04 |
| LOM | 1.18 | 0.20 | 0.91 | **0.95** | 19.59 | 1.38 | 0.29 | 0.88 | **0.94** | 19.88 | 2.44 | 0.30 | 0.82 | **0.94** | 20.04 |
| $L_p$ | 2.98 | 0.52 | 0.75 | 0.85 | 82.90 | 3.76 | **0.54** | 0.77 | 0.84 | 97.81 | 4.99 | 0.52 | 0.69 | 0.85 | 66.98 |
| **XAIFOOLER** | **3.95** | **0.62** | **0.73** | 0.88 | 47.49 | **4.75** | 0.54 | **0.74** | 0.89 | 48.76 | **6.11** | **0.62** | **0.67** | 0.89 | 38.79 |
| **GB** Inherency | 0.53 | 0.16 | 0.89 | 1.00 | 167.35 | 0.55 | 0.15 | 0.87 | 1.00 | 171.88 | 0.44 | 0.16 | 0.93 | 1.00 | 169.19 |
| Random | 1.31 | 0.32 | 0.72 | 0.81 | 618.18 | 1.38 | 0.30 | 0.75 | 0.81 | 616.60 | 0.99 | 0.23 | 0.79 | 0.82 | 637.65 |
| LOM | 0.60 | 0.22 | 0.89 | **0.91** | 322.85 | 0.55 | 0.11 | 0.90 | **0.91** | 312.81 | 0.47 | 0.10 | 0.91 | **0.91** | 295.33 |
| $L_p$ | **2.06** | 0.39 | **0.66** | 0.86 | 583.15 | 1.99 | 0.47 | 0.71 | 0.86 | 547.91 | 1.47 | 0.43 | **0.77** | 0.87 | 553.80 |
| **XAIFOOLER** | 2.02 | **0.48** | 0.71 | 0.89 | 358.88 | **2.10** | **0.52** | 0.71 | 0.89 | 368.53 | **1.56** | **0.45** | 0.78 | **0.91** | 353.98 |

**bold** and underline statistics denote the best and second best results except "Inherency"

Table 3: Experiment results in terms of explanation changes (ABS, RC, INS) and semantic preservation (SIM, PPL)

the set-based approaches while retaining the positional information and capacity for feature weightings. The weights assigned to the features are controlled by a convergent series where the proportion of weights associated with the first $k$ features is determined by a hyper-parameter $p \in [0, 1]$. As $p \to 0$ more weight is assigned to the topmost features, while as $p \to 1$ the weight becomes distributed more evenly across all possible features.

Fig. 3 illustrates the comparison between RBO, Jaccard, and Kendall/Spearman measures, which highlights two advantages of RBO. First, RBO allows features outside the top $k$ some small influence on the resulting similarity. Second, the weighting scheme associated with RBO allows more granularity when determining similarity by applying weight to the top $k$ features (Fig. 3), which is lacking with the other measures. In other words, RBO provides richer signals to guide the greedy optimization step of XAIFOOLER. Moreover, RBO allows us to decide how much distribution mass to assign to the top-$k$ features (Fig. 3 with 90% & 80% mass)). This enables us to focus on manipulating the top-$k$ features that the end-users mostly care about while not totally ignoring the remaining features. We refer the readers to Appendix B for detailed analysis on RBO and other measures.

## 6 Experiments

### 6.1 Set-up

**Datasets and Target Models.** We experiment with three datasets: sentiment analysis (IMDB) (Maas et al., 2011), symptom to diagnosis classification (S2D) (from Kaggle) and gender bias classification (GB) (Dinan et al., 2020), of varying averaged lengths (230, 29, 11 tokens), and number of labels (2, 21, 2 labels) (see Table B.7-Appendix). Each dataset is split into 80% training and 20% test sets. We use the training set to train three target models, namely DistilBERT (Sanh et al., 2019), BERT (Devlin et al., 2019) and RoBERTA (Liu et al., 2019), achieving around 90%–97% in test prediction F1 score.

**Baselines.** We compare XAIFOOLER against four baselines, namely (1) *Inherency*: the inherent instability of LIME due to random synthesis of training examples to train the local surrogate model (Sec. 3); (2) *Random*: randomly selects a word to perturb and also randomly selects its replacement from a list of nearest neighbors (3) *Location of Mass (LOM)*: inspired by Sinha et al. (2021), similar to XAIFOOLER but uses COM (Sec. 5.3) as the explanation similarity function; (4) $L_p$: similar to XAIFOOLER but uses $L_2$ (Sec. 5.3) as the explanation similarity function.

**Evaluation Metrics.** Similar to prior works, we report the explanation changes of top-$k$ features using: *Absolute Change* in ranking orders (ABS↑); *Rank Correlation* (RC↑), which is calculated by the formula: $1 - max(0, \text{Spearman-Correlation}(e_{d_b}[:k], e_{d_p}[:k])$; Intersection Ratio (INS↓)–i.e., ratio of features remained in top $k$ after perturbation. Moreover, we also report the semantic similarity between $d_b$ and $d_p$ (SIM↑) by calculating the cosine-similarity of their vectors embedded using USE (Cer et al., 2018); and the naturalness of $d_p$ by calculating its perplexity score (PPL↓) using large language model GPT2-Large (Radford et al., 2019)

| Method | Average across all Results | | | | |
|---|---|---|---|---|---|
| | ABS↑ | SM↑ | INS↓ | SIM↑ | PPL↓ |
| Rnd Order+Rnd Search | 3.51 | 0.40 | 0.79 | 0.85 | 259.23 |
| Rnd Order+Greedy Search | 4.63 | 0.51 | 0.79 | **0.91** | **123.45** |
| Greedy Order+Greedy Search | **6.11** | **0.59** | **0.73** | 0.89 | 157.15 |

"Greedy Order+Greedy Search": Ours; "Rnd": Random

Table 4: Both step 1 (Greedy Order) and step 2 (Greedy Search) (Sec. 5.1) are crucial to XAIFOOLER.

as a proxy. Arrow ↑ and ↓ denote *the higher, the better* and *the lower, the better*, respectively.

**Implementation Details.** We set $\epsilon \leftarrow 0.1$ with a minimum allowance of 3 tokens to account for short texts. We use Fig. B.6 to select the hyper-parameter $p$ values of RBO that correspond to 90%, 95%, 95% mass concentrating in the top-5 ($k \leftarrow 5$), top-3 ($k \leftarrow 3$) and top-2 ($k \leftarrow 2$) features for IMDB, S2D and GB dataset, respectively. Based on Sec. 3, we set the sampling rate $n$ such that it is sufficiently large to maintain a stable change in LIME's inherent instability, resulting in $n$ of 4.5K, 2.5K and 1.5K for IMDB, S2D and GB dataset. During perturbation, we constrain the final perturbed document to result in at least one of the top $k$ features decreasing in rank to mitigate sub-optimal perturbations being accepted ultimately increasing the quality of the final perturbed document by allowing us to set a lower threshold for the total number of perturbations at the cost of more computational effort. Appendix Sec. A.1 includes the full reproducibility details.

### 6.2 Results

**Overall.** LIME is unstable and vulnerable against text perturbations. Table 3 shows that XAIFOOLER significantly outperformed all baselines by large margins in its ability to manipulate LIME's explanations, showing average changes in $\Delta$ABS↑, $\Delta$RC↑ and $\Delta$INS↓ of +128%, +101% and -14.63% compared to *Inherent Instability*, and of +22%, +16% and -3.15% compared to the 2nd best $L_p$ baseline. Moreover, it also delivered competitive results for semantic preservation, consistently outperforming all the baselines except *LOM*, which showed to be very good at preserving the original semantics. This happened because LOM scores often reached 0.5–i.e., moving the location of centered mass to over 50% of total length, very quickly without many perturbations, yet this did not always guarantee actual changes in feature rankings.

**Ablation Test.** Table 3 confirms the appropriate-

| Ranking Changes Before and After Perturbation |
|---|
| "@user sick of liberals thinking it's ok to dictate where they ~~think~~ **thinking** israeli jews should be allowed to live, including in israel" |
| *liberals*: 1st → 2nd; *israeli*: 3nd → 4rd; *thinking*: 5th → 1st; |

Table 5: Case Study: Perturbation Examples and the Corresponding Changes in Explainable Feature Rankings on Twitter Hate Speech Detection.

| Model/ | DistilBERT | | BERT | | RoBERTA | |
|---|---|---|---|---|---|---|
| Dataset | Doc | Token | Doc | Token | Doc | Token |
| IMDB | 331.5 | 9.1 | 505.2 | 13.9 | 564.9 | 15.5 |
| S2D | 242.5 | 12.0 | 518.5 | 25.7 | 482.1 | 23.9 |
| GB | 188.7 | 18.2 | 385.7 | 36.9 | 421.1 | 40.6 |

Table 6: Average attack runtimes in seconds per document and token (NVIDIA RTX A4000).

ness of using RBO over other similarity function such as $L_p$ and *LOM* (Sec. 5.4). Evidently, replacing any of the procedure steps of XAIFOOLER with a *random* mechanism dropped its performance (Table 4). This demonstrates the importance of both Step 1 (greedy ordering of tokens to decide which one to perturb first) and Step 2 (greedy search for best perturbations) described in Sec. 5.1.

## 7 Discussion

**Case Study: XAI and Content Moderation.** XAI systems are often used in conjunction with human-in-the-loop tasks such as online content moderation–e.g., detecting hate speech or fake news, where quality assurance is paramount. Thus, manipulations on such XAI systems will lead to human experts receiving incorrect signals and hence resulting in sub-optimal results or even serious consequences. Evaluation of XAIFOOLER on 300 positive examples from a Twitter hate speech detection dataset with $k \leftarrow 2$ shows that XAIFOOLER often demotes the 1st-rank feature out of top-2, reducing *RC* to 0.4 and INS to 0.78, while still maintaining the original semantics with *SIM*>0.9. This means XAIFOOLER can force the AI system to output *correct predictions but for wrong* reasons. Table 5 describes such a case where XAIFOOLER uses the existing "thinking" to replace "think", promoting this feature to 1st rank while demoting more relevant features such as "liberals" and "israeli".

**Instability Distribution.** Fig. 4 illustrates the distributions of explanation shifts after perturbations on IMDB (S2D, GB results are included in the Appendix). This shows that LIME's instability expresses differently among documents, with some

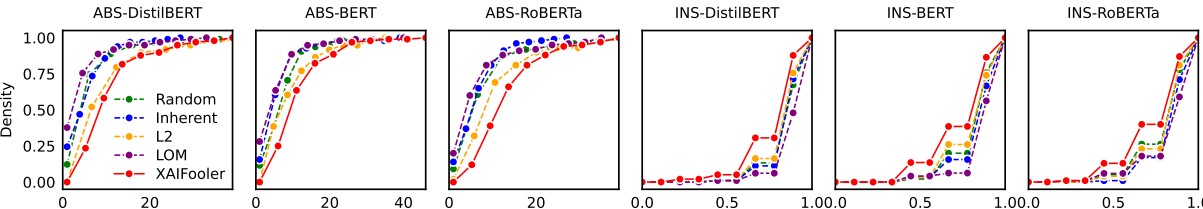

Figure 4: CDF plots of absolute change (ABS↑) and $k$-Intersection (INS↓) statistics on IMDB dataset.

are easier to manipulate than others. Fig. 4 also shows that if one to calculate attack success rate using a threshold, which is often customized to specific applications, XAIFOOLER will still consistently outperform the baselines.

**Document Length v.s. Sampling Rate v.s. Runtime.** The majority of computation time is spent generating explanations using LIME. Table 6 shows that the average runtime slowly increases as the document length grows while the time per token decreases. Moreover, larger documents have more possibilities for effective perturbations, resulting in a more efficient search and ultimately fewer explanations to generate. Shorter documents are faster to run LIME, but require extensive searching to find acceptable perturbations and so generate many explanations. Furthermore, for shorter documents the explanations are quite stable even at rates much lower than the default, and as the document size increases, lower sampling rate values begin to degrade stability (Fig. 2, Table 1).

## 8 Conclusion

This paper affirms that LIME is inherently unstable and its explanations can be exploited via text perturbations. We re-purpose and apply RBO as a key part of our algorithm XAIFOOLER, which we have shown to provide competitive results against other adversarial explanation generation methods, and do so in a way that satisfies our criteria from earlier when determining just how best to compare an explanation. Future works include understanding the interactions between the target model and LIME that lead to such instability.

## Limitations

Our method and the results collected from it used the default settings and parameters for the explanation generation framework within TEXTEX-PLAINER[*] with the exception of the sampling rate $n$. This departure is justified by the results shown

---

[*]https://eli5.readthedocs.io/

in Table 1 and especially Fig. 2, where we see a rapid diminishing returns in explanation fidelity as the sample rate $n$ increases. Although our choice of sampling rates are already significantly higher than prior works, this reduction in sampling rate was necessary due to the extensive computation time required, with the explanation generating process being the largest proportion of computation time. The extensive computation time prevented extensive testing with a large range of sampling rates, especially those higher than the default of $n=5K$. We leave the question of what effect, if any, "oversampling" with rates much higher than the default has on stability to future work. It remains unexplored just how much effect the base model itself has on the resulting explanation similarity. That is, are certain type of models amplifying LIME's unstable results? This question also applies to the LIME itself. Our choice of surrogate model was the default, but this can be replaced. What are the differences between the choices for local model when it comes to stability?

## Broader Impacts and Ethics Statement

The authors anticipate there to be no reasonable cause to believe use of this work would result in harm, both direct or implicit. Similar to existing adversarial attack works in NLP literature, there might be a possibility that malicious actors might utilize this work to manipulate real life AI systems. However, we believe that the merits of this work as in providing insights on LIME's stability outweigh such possibility. Concerns about computational expenditure, and the remedies taken to reduce impact, are addressed in Sec. A.1.3. The authors disclaim any conflicts of interest pertaining to this work.

## Acknowledgements

L. Chen's work is partially supported by the NSF under grant CNS-2245968. The authors thank the anonymous reviewers for their many insightful comments and suggestions.

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

## A  Reproducibility

### A.1  Implementation Details

This section provides all details needed to reproduce our experiment results. The open-source implementation is available at: `https://github.com/cburgerOlemiss/XAIFooler`

#### A.1.1  Hardware

Facts and figure relating to computation time were generated using a single NVIDIA RTX A4000 on a virtual workstation with 45 GiB RAM.

#### A.1.2  Frameworks / Software

We use an off-the-shelf LIME implementation, TEXTEXPLAINER, itself a subset of the popular explainability package ELI5[†]. TEXTEXPLAINER provides extensive functionality for formatting and printing explanation output. XAIFOOLER is implemented using the TEXTATTACK framework (Morris et al., 2020), a full featured package designed for adversarial attacks in natural language. TEXTATTACK's design offers the ability to quickly extend or implement most aspects of the adversarial example generation workflow.

#### A.1.3  Parameter Selection

Our values for the RBO weighting were 0.75, 0.49, and 0.32 respectively for the IMDB, S2D, and GB datasets. These values were determined based on the attributes in Sec. 5.2 and the average length of the explanation features. Shorter document lengths generate smaller explanations, and so have more importance associated with the smaller number of features. We must treat altering the positions of the top 5 features in an explanation with 7 total features as more significant than for an explanation with 150 features. This significance is controlled by the RBO value with a smaller value assigning more importance to the top features.

Our values for the sampling rates were 4,500, 2,500, and 1,500 respective for the IMDB, S2D, and GB datasets. These values were derived based on the analysis shown in Fig. 2 where the baselines for the inherent stability were determined. Our goal is to balance explanation fidelity and running time, as is seen in Table 6 the computational expenditure for this process is extensive.

The maximum perturbation budget was set to 20% of the total document length (rounded up as to guarantee at least one perturbation). This budget

---

[†]`https://eli5.readthedocs.io/`

is lower than some other work in adversarial explanations, but the requirement for consistency in the meaning of the perturbed document necessitated a modest limit to the perturbations. This consistency, especially for larger documents, began to degrade appreciably beyond $\sim20\%$.

## B  Motivating RBO

Our use of RBO follows from our choice of desired attributes in Sec. 5.2. Here we provide examples to justify our selection of similarity measure expanding on the summary in Table 2. Note that the weight-based distances $L_p$, and $LOM$ are different enough from the structure of the feature-based measures like RBO that direct comparison is difficult. Their construction gives them particular strengths but they fundamentally are unable to satisfy our desired attributes. They remain included as important metrics used for comparison against the other similarity measures.

However, RBO can be compared directly to the Jaccard and Kendall / Spearman similarities. We do so in Fig. 3 where we demonstrate two of RBO's advantages. First, RBO provides some capacity for determining similarity when using features not within the selected top $k$. Second, the weighting scheme associated with RBO allows more granularity when determining similarity; in other words, the similarity output is smoother due to the weight assignment, which is lacking with the other measures. Here a fixed list of 50 features is generated, which is then uniformly shuffled 50,000 times. The results are plotted according to the similarity output from each of the measures with respect to the top 5 features. We see RBO allows some small amount of similarity to remain desipte assigning the significant majority of the mass to the top 5 features. The other measures return 0 similarity for over half the permutations, as our requirement for a concise explanation allows us only to concentrate on the top few features. The other measures show poorer delineation between important features (as determined by position in the top 5), resulting in the step-wise behavior (explained more in the measure's respective subsections below).

### B.1  Jaccard Index

The Jaccard Index, being entirely set-based, preserves no order between two lists of feature explanations. For collections of features $F_1 = [a, b, c]$ and $F_2 = [c, b, a]$ the Jaccard Index

of $F_1$ and $F_2$ J($F_1,F_2$) = 1 despite no pair of values having an equal position. These lists are clearly different, but the Jaccard Index does not have the capacity to differentiate between them. Additionally, the Jaccard Index assigns an equal value, related to the total number of elements, for any dissimilarity between the lists. For example, let

$$F_1 = [a, b, c, ..., x, y, z]$$
$$F_2 = [\alpha, b, c, ..., x, y, z]$$
$$F_3 = [a, b, c, ..., x, y, \omega]$$

With $F_1$ being ordered by weight in decreasing importance.

Finally let weight($\alpha$) < weight($a$) and weight($y$) > weight($\omega$) > weight($z$).

Then we have J($F_1$, $F_2$) = J($F_1$, $F_3$).

Clearly the similarity between $F_1$ and $F_2$ should be considered less than $F_1$ and $F_3$ as we have additional information pertaining to each feature's importance.

### B.2  Kendall's $\tau$ & Spearman's $\rho$

Kendall's $\tau$ & Spearman's $\rho$ are subject to similar issues affecting the Jaccard Index. Both $\tau$ and $\rho$ can handle the first scenario, while the Jaccard Index could not. That is, for $F_1 = [a, b, c]$ and $F_2 = [c, b, a]$ then Kendall($F_1,F_2$) $\neq$ 1 and Spearman($F_1,F_2$) $\neq$ 1. However, $\tau$ and $\rho$ are similar to Jaccard in that the difference between weighted features has not been taken into account. For example, let

$$F_1 = [a, b, c, ..., x, y, z]$$
$$F_2 = [b, a, c, ..., x, y, z]$$
$$F_3 = [a, b, c, ..., x, z, y]$$

Then Kendall($F_1$, $F_2$) = Kendall($F_1$, $F_3$), and Spearman($F_1$, $F_2$) = Spearman($F_1$, $F_3$).

The exchange between features $a$ and $b$ should be considered more important than the exchange between features $y$ and $z$. The resulting value is also related to the size of the lists, and a single exchange of two adjacent becomes less important as the size of the lists grows larger.

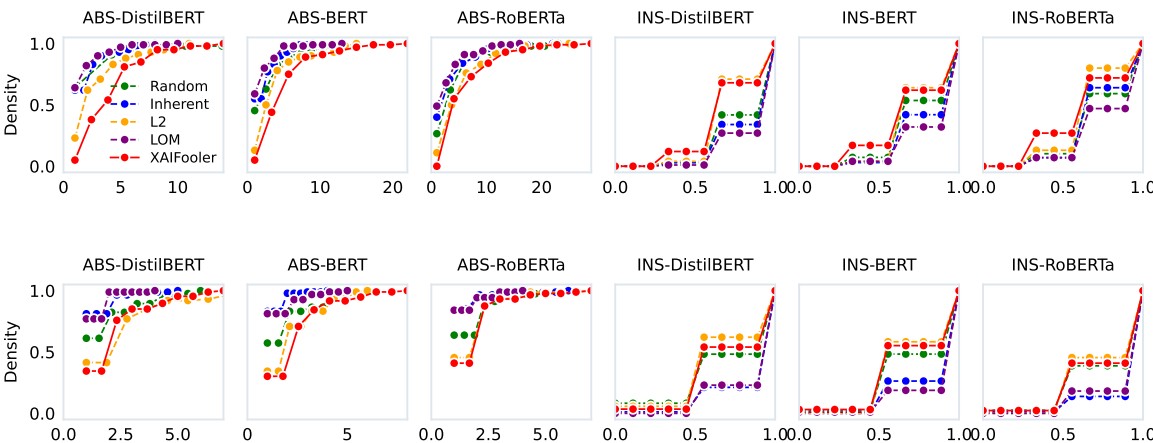

Figure B.5: CDF plots of absolute change (ABS↑) and $k$-Intersection (INS↓) statistics on S2D (top) and GB (bottom) dataset.

## B.3 $L_p$

$L_p$ discards positional importance and instead concentrates on the weights associated with the features. We can encounter a scenario where the order of the features is identical, but the weights have shifted enough to provide a substantial $L_p$ distance. Consider a list of features $F_1 = [a, b, c]$ and $F_2 = [a, b, c]$ with weights $W_1 = [3, 2, 1]$ and $W_2 = [4, 3, 2]$. Clearly the $L_p$ distance here will be non-zero, but the position of the features remains identical. Does the change in total weight matter? It may, and $L_p$ is well specified if that is the case. However, we may not necessarily understand just how important the differences between weights are. Because of this, we do not wish to set a threshold for significance between the two explanations purely on a distance between the weights. We instead appeal to the position of the features, a coarse form of weighting, and use our own scheme for weight importance that concentrates the weights in an easily tuned manner of our own choosing.

## B.4 Location of Mass

While $L_p$ determines that weight is shifting, LOM provides insight into where it is moving. If the location (usually center) of the cumulative weights has shifted, then more weight has been assigned to or away from certain features and so the explanation must then be different. LOM has issues similar to $L_p$ in that the actual location of mass may not shift despite weight being changed significantly among many features. For example, consider the lists of features

| Dataset | # Tokens | # of Labels | DistilBERT | BERT | RoBERTa |
|---------|----------|-------------|------------|------|---------|
| IMDB | 230 | 2 | 0.91 | 0.92 | 0.94 |
| S2D | 29 | 21 | 0.94 | 0.93 | 0.97 |
| GB | 11 | 2 | 0.90 | 0.90 | 0.89 |

Table B.7: Dataset statistics and prediction performance (F1 Score) of target classification models on test set.

$$F_1 = [a, b, c, d, e]$$
$$F_2 = [a, b, c, d, e]$$

With weights $W_1 = [1, 0, 5, 0, 1]$
$W_2 = [1.3, 1.2, 2, 1.2, 1.3]$

Both collections of features possess the same total weight, and the same location of mass (center). But $F_2$ has a significantly different assignment of weights and it is reasonable to conclude that this explanation is different.

Additionally, for certain explanations that are highly concentrated in weight it may not be feasible to find perturbations that are able to distribute this among other features to the extent necessary to move the location of mass. For example, let $W_3 = [1, 2, 1, 1000, 2, 2, 1]$. We would have to redistribute the overwhelming majority of the weight to the other features in order to move this location of mass away from index 3.

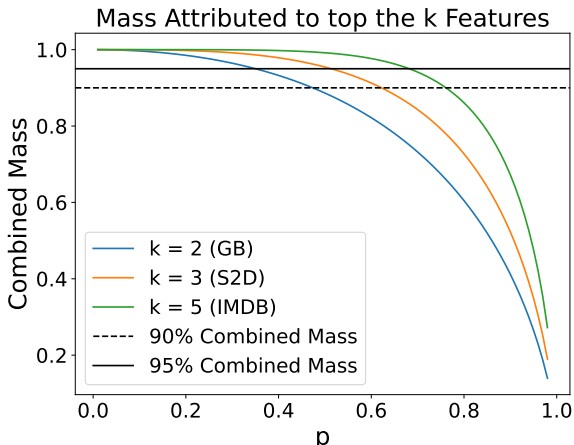

Figure B.6: Mass associated with the top $k$ features for different values of RBO's hyper-parameter $p$.

## C    Other Supplementary Materials

1. Statistics of experimental datasets and the performance of target models on their test sets (Table B.7)

2. Relationship between combined distribution mass, $k$ and hyper-parameter $p$ of RBO (Fig. B.6)

3. CDF plots for GB and S2D Datasets (Fig. B.5)