# OpenReview forum: ""Are Your Explanations Reliable?" Investigating the Stability of LIME in Explaining Text Classifiers by Marrying XAI and Adversarial Attack"
_EMNLP/2023/Conference — EMNLP 2023 Main_

### Official Review · Reviewer_pyA2 · 2023-08-04

**Soundness:** 3

**Excitement:**

2: Mediocre: This paper makes marginal contributions (vs non-contemporaneous work), so I would rather not see it in the conference.

**Paper Topic And Main Contributions:**

This paper first show that LIME is inherently unstable and produces inconsistent results, especially in low sampling rates. Moreover, they propose a word-replacement-based attack, XAIFOOLER, that perturbs text inputs and manipulate explanations generated by LIME. They define several constraints to maintain the similarity between the adversarial and original document while reducing the similarity between the explanation of the original and adversarial document. They propose to use Rank biased Overlap (RBO) to measure the similarity between two explanations.

**Questions For The Authors:**

- Question A: What are the main differences between this work and Location of Mass (Sinha et al., 2021)?
- Question B: What is the necessity of constraint of eq (5)?
- Question C: Please provide some of the examples generated by XAIFOOLER.


**Reasons To Accept:**

- The paper is well-written and clear.
- The study of the inherent instability of LIME is interesting.
- The experiments are thorough, and the attack outperforms the baselines.

**Reasons To Reject:**

- The method is very similar to the method of Location of Mass (Sinha et al., 2021). It seems that the only difference is the metric used to find the similarity between two explanations (RBO vs COM) and the larger sampling rate. This makes the contribution of the paper limited.

- It is not clear why the k important features must appear in the perturbed explanation (eq (5)). It is necessary that the top k features not be perturbed by the attack, but it does not seem necessary that they appear in the perturbed explanation.

**Reproducibility:**

5: Could easily reproduce the results.

**Reviewer Confidence:**

4: Quite sure. I tried to check the important points carefully. It's unlikely, though conceivable, that I missed something that should affect my ratings.

---

> ### Author Rebuttal · Authors · 2023-08-29
>
> **Summary:** The authors would like to thank the reviewer for their valueable time and feedback. We respectfully ask the reviewer to reconsider our work's novelty beyond the baseline Location of Mass (Since et al., 2021) as we proposed a very different experiment setting and constraints. We would like to address all of the concerns in detail as follows.
>
> > Differences from Location of Mass (Sinha et al., 2021)
>
> We view our work as complementary to the Location of Mass paper (Sinha et al., 2021) in that by establishing a starting point for investigation through an understanding of the inherent instability other strategies for measuring the robustness of XAI methods can result in more *meaningful conclusions*. We consider our contributions to be threefold:
>
> **(1) An exploration of the inherent instability within LIME:** Determining levels of instability within LIME before any perturbations are applied is necessary to produce meaningful results in future work. By establishing such data we can provide a way to prevent inappropriate conclusions from being drawn to due sampling bias. This strengthens future research efforts in that more reasonable judgments can be made about LIME’s efficacy. We also want to emphasize that our sampling rate is not only set as a much larger value (e.g., 9x on IMDB dataset) compared to Location of Mass (Sinha et al., 2021) and hence a more challenging experiment setting, the introduction of the inherent instability analysis also provides reasonings on which minimally sufficient sampling rate to be used.
>
> **(2) A systematic motivation on the proper selection of similarity measure for adversarial XAI:** As the similarity measure is the heart of the comparison between explanations, it is directly responsible for the conclusions drawn from the experiments. By motivating specific requirements based on both the general adversarial XAI process, and the individual particulars of LIME itself we can again draw more systematic conclusions. Thus, the provided insights and reasons on how RBO is different from existing metrics such as LOM and L_p is not merely a random adoption, and serves as a framework for meaningful discussion in future works.
>
> **(3) A meaningful search methodology and practical threat model:** Different from Location of Mass (Sinha et al., 2021) which allows perturbations of important, top-ranked explanation features/words, our choice to perturb the unimportant words initially gives us the advantage of casting the adversarial process directly and practically in terms of the definition of stability where small input changes should result in small output changes in real life. We do consider the LOM measure as interesting and underutilized with respect to text data. And especially well suited for addressing the similar research question where the most important features are selected for perturbation, testing the stability in terms expecting significant changes in the resulting explanation for changes in the important feature(s). We instead elected to pursue the search for perturbations of unimportant features in contrast to some related work to demonstrate this instability in a way that is analogous to traditional adversarial attack on text models with subtle perturbations. This subtlety we felt would be best achieved by avoiding perturbations of important words in the explanation due to the large changes in explanation similarity despite otherwise quality substitutions.
>
> > Necessity of constraint of eq (5)
>
> We think that our descriptions of Eq. (5) might come across as a misunderstanding. Our intent behind Eq. (5) is only to prevent the perturbation of features that are considered significantly important by LIME. Our primary concern is that if they are candidates for perturbation, the results become uninteresting in that changing an important feature results in an appreciable change in the explanation. We are unconcerned with the location of this feature, just that it remains within the explanation. Hence, Eq. (5) actually makes our problem more challenging and practical, hence worth investigating.
>
> > Some of the examples generated by XAIFOOLER
>
> Both figure 1 and table 5 are actual output of the process, with the blue text the original word(s) and the red text the chosen substitutions. We have included one of them in the table below from the symptom to diagnosis classification (S2D) dataset and Twitter hate-speech detection dataset.
>
> | Original Text | Perturbed Text | Original Features (Ordered by ranks) | Perturbed Features (Ordered by ranks) |
> |---|---|---|---|
> | I have a skin rash that gets worse in the winter. I have to moisturize more regularly and have [...] I also have joint pain | I have a skin rash that becomes worse in the winter. I have to moisturize more consistently and have [...] I also have joint pain | skin winter joint moisturize regularly worse gets air dry use moisturized painrash humidifiers | skin rash worse pain winter moisturized moisturize dry humidifiers consistently joint use air |
> | @user sick of liberals thinking it's ok to dictate where they think israeli jews should be allowed to live, including in isrâ | @user sick of liberals thinking it's ok to dictate where they thinking israeli jews should be allowed to live, including in isrâ | liberals isrâ israeli jews thinking allowed | thinking liberals isrâ israeli jews dictate |

---

### Official Review · Reviewer_bxaW · 2023-08-06

**Soundness:** 4

**Excitement:**

4: Strong: This paper deepens the understanding of some phenomenon or lowers the barriers to an existing research direction.

**Paper Topic And Main Contributions:**

This paper's contributions are twofold. First, this paper demonstrates the fundamental instability of a popular explainability technique, LIME (Ribeiro et al., 2016), which displays inconsistent explanation results on textual experiments just by varying the sampling rate (the number of local training examples used to train its surrogate model), without alterations to input or model parameters.

Second, this paper additionally contributes an adversarial attack algorithm, XAIFooler, which can distort the explanations produced by LIME via minimal text perturbations. Formulated as an adversarial text attack optimization problem, this algorithm takes in an input document and conducts an iterative greedy word selection / search process, ultimately returning a perturbed document that fulfills a list of constraints: 1). its explanations should be maximally different, 2). the original prediction is preserved, 3). there is sufficiently large semantic similarity between the base and perturbed documents, 4). the number of possible perturbations is constrained, 5). the top-k ranked features from the base explanation likewise also appear in the perturbed explanation.

The paper also makes a point to identify a good explanation similarity function, which should account for positional importance, the continuous weights of feature importance scores, disjoint features, and unequal list lengths. The paper identifies Rank-based Overlap (RBO) as a suitable candidate among several other measures. Across three textual classification tasks and three BERT-based models, XAIFooler displays sufficiently convincing results at 1). distorting the explanation feature ranking, 2). maintaining adequate similarity between old and perturbed documents, 4). maintaining naturalness of text via the perplexity.

**Questions For The Authors:**

A. Is it possible to provide examples of base and perturbed documents?

B. Do you think XAIFooler can be applied to other XAI methods besides LIME, e.g., SHAP or GradCAM?

**Reasons To Accept:**

This is a well-written paper, and easy to follow. The algorithm is well-articulated, and framing it as an adversarial text generation optimization problem makes sense. I appreciate the careful baseline comparisons in Table 3, as well as the extensive ablations which show (1) that RBO is a well-suited similarity function, and (2) both greedy ordering and greedy search are integral to XAIFooler's good performance. Additionally, the case study in the Discussion section on content moderation underlines the importance of building AI systems that are robust against adversarial input produced by the likes of XAIFooler.

Overall, minor changes aside, I like this paper and would like to see it accepted for publication.

**Reasons To Reject:**

One weakness to this paper is that it is not clear to me the linkage between the fundamental instability of LIME, and how XAIFooler is meant to affect this instability. I'm not really convinced by the justification in L246-L251, since it seems like we are talking about two different issues here: LIME produces inconsistent results when the sampling rate is varied, and LIME can further be induced to produce inconsistent results when provided a minimally perturbed document. I think a more in-depth explanation or re-framing is in order here.

**Reproducibility:**

4: Could mostly reproduce the results, but there may be some variation because of sample variance or minor variations in their interpretation of the protocol or method.

**Reviewer Confidence:**

2: Willing to defend my evaluation, but it is fairly likely that I missed some details, didn't understand some central points, or can't be sure about the novelty of the work.

**Typos Grammar Style And Presentation Improvements:**

Grammar:

L434: "(Kumar and Vassilvitskii, 2010)" -> "Kumar and Vassilvitskii, 2010"

L492: "inspired by (Sinha et al., 2021)" -> "inspired by Sinha et al., 2021"

---

> ### Author Rebuttal · Authors · 2023-08-29
>
> **Summary:** The authors would like to thank the reviewer for their time and especially for their cogent summary of the paper. We would like to address the concerns listed below.
>
> > Connection between LIME’s instability and XAIFooler perturbation method
>
> The initial instability exploration resulted from questions about the effectiveness of early forms of XAIFooler. In related works we saw a lack of discussion on what levels of inherent instability imparted due to LIME’s sampling process within the explanation generation. In order to avoid reporting seemingly significant results that fell within whatever baseline instability already exists (For example, in Figure 2 if a perturbed document’s similarity under a comparable measure returned ~80%, we would not be able to make an effective judgment that it was unstable due to the baseline being ~75% on an unperturbed document), we sought to establish a reasonable minimum sample rate that balanced explanation convergence and performance. We consider XAIFooler as an extension of the exploration of the instability associated with LIME (or other similar methods, see our more specific response to your question about generalizability of XAIFooler to other XAI methods) beyond the initial instability resulting from the sampling process. The authors would gladly clarify the connection between XAIFooler itself and the initial instability discussion as a revision.
>
> >  Is it possible to provide examples of base and perturbed documents?
>
> Both figure 1 and table 5 are actual output of the process, with the blue text the original word(s) and the red text the chosen substitutions. We have included one of them in the table below from the symptom to diagnosis classification (S2D) dataset and Twitter hate-speech detection dataset.
>
> | Original Text | Perturbed Text | Original Features (Ordered by ranks) | Perturbed Features (Ordered by ranks) |
> |---|---|---|---|
> | I have a skin rash that gets worse in the winter. I have to moisturize more regularly and have [...] I also have joint pain | I have a skin rash that becomes worse in the winter. I have to moisturize more consistently and have [...] I also have joint pain | skin winter joint moisturize regularly worse gets air dry use moisturized painrash humidifiers | skin rash worse pain winter moisturized moisturize dry humidifiers consistently joint use air |
> | @user sick of liberals thinking it's ok to dictate where they think israeli jews should be allowed to live, including in isrâ | @user sick of liberals thinking it's ok to dictate where they thinking israeli jews should be allowed to live, including in isrâ | liberals isrâ israeli jews thinking allowed | thinking liberals isrâ israeli jews dictate |
>
> > Do you think XAIFooler can be applied to other XAI methods besides LIME, e.g., SHAP or GradCAM?
>
> XAIFooler can be used directly with any post-hoc XAI method that returns a tuple of features such as LIME or SHAP. For methods that lack a relative ordering of the feature importance XAIFooler is compatible but requires an appropriate choice of similarity measure, namely for RBO the weight associated with each feature must be considered equal which can be adjusted simply by altering RBO’s dedicated weighting parameter.
>
> By enabling XAIFooler to accept white-box classification model, it will be compatible with other XAI methods that are gradient based such as GradCAM. However, we are uncertain if we will observe similar trends in the instability results. The authors consider it an important question, both in the specific case for XAIFooler directly, and in the general case where the entire corpus of related work in text stability can generalize to gradient based methods.

---

### Official Review · Reviewer_BX5w · 2023-08-08

**Soundness:** 3

**Ethical Concerns:**

Yes

**Excitement:**

2: Mediocre: This paper makes marginal contributions (vs non-contemporaneous work), so I would rather not see it in the conference.

**Justification For Ethical Concerns:**

Minor: the work describes attacks on model explanations but offers no prescriptions how to prevent
or defend against them. It is unbalanced towards the negative side of the societal impacts scale.

**Paper Topic And Main Contributions:**

The work describes the evaluation of the "stability" of explanations of input importance in NLP
models as computed by the LIME method. The method involves making small perturbations in the inputs
so as to preserve their meaning but to change the explanations in terms of producing different
rankings of most important input words. Various metrics are tested for both preservation of meaning
during perturbation and the resulting explanation dissimilarity. Compared to several baselines, the
presented work termed "XAIFooler" does mostly better save for various points where it is beat in
terms of input semantic preservation by one baseline method. The paper uses these results to both
demonstrate the better performance of their perturbation method and to argue that LIME explanations
are inherently unstable.

**Questions For The Authors:**

- Question A: The Disjoint Features requirement seems to be due to simplify your implementation and
  not due to some desirable property. Is it?

  Possibly related, the example Table 5 has no "think" in the perturbed output. What is thus the
  ranking of "think" in that perturbation?

- Question B: Table 3 has a lot of metrics shown. I imagine each of the metrics can be made an
  optimization goal in an explanation attack which would then give it an advantage in terms of that
  metric at least. Can you adjust your method at least with different target goals and see whether
  they are independently optimizable?

**Reasons To Accept:**

+ Proposed method has overall better performance at least in terms of the metrics tested.

+ Several baselines compared experimentally.

**Reasons To Reject:**

- While studying instability of LIME, the work likely confuses that instability with various
  other sources of instability involved in the methods:

  - Instability in the model being explained.

  - Instability of ranking metrics used.

  - Instability of the LIME implementation used. This one drops entire words instead of perturbing
    embeddings which would be expected to be more stable. Dropping words is a discrete and
    impactful process compared to embedding perturbation.

  Suggestion: control for other sources of instability. That is, measure and compare model
  instability vs. resulting LIME instability; measure and compare metric instability vs. resulting
  LIME instability. Lastly consider evaluating the more continuous version of input perturbation
  for LIME based on embeddings. While the official implementation does not use embeddings, it
  shouldn't be too hard to adjust it given token embedding inputs.

- Sample complexity of the learning LIME uses to produce explanations is not discussed. LIME
  attempts to fit a linear model onto a set of inputs of a model which is likely not linear. Even
  if the target model was linear, LIME would need as many samples as there are input features to be
  able to reconstruct the linear model == explanation. Add to this likelihood of the target not
  being linear, the number of samples needed to estimate some stable approximation increases
  greatly. None of the sampling rates discussed in the paper is suggestive of even getting close to
  the number of samples needed for NLP models.

  Suggestion: Investigate and discuss sample complexity for the type of linear models LIME uses as
  there may be even tight bounds on how many samples are needed to achieve close to optimal/stable
  solution.

  Suggestion: The limitations discusses the computational effort is a bottleneck in using larger
  sample sizes. I thus suggest investigating smaller models. It is not clear that using
  "state-of-the-art" models is necessary to make the points the paper is attempting to make.

- Discussions around focus on changing or not changing the top feature are inconsistent throughout
  the work and the ultimate reason behind them is hard to discern. Requiring that the top feature
  does not change seems like a strange requirement. Users might not even look at the features below
  the top one so attacking them might be irrelevant in terms of confusing user understanding.

       "Moreover, its experiment settings are not ideal as it allows perturbations of top-ranked
        predictive features, which naturally change the resulting explanations"

  Isn't changing the explanations the whole point in testing explanation robustness? You also cite
  this point later in the paper:

       "Moreover, this requirement also accounts the fact that end-users often consider only the
        top k most important and not all of the features"

  Use of the ABS metric which focuses on the top-k only also is suggestive of the importance of top
  features. If top features are important, isn't the very top is the most important of all?

  Later:

       "... changing the most important features will likely result in a violation to constraint in
        Eq. (2)"

  Possibly but that is what makes the perturbation/attack problem challenging. The text that
  follows does not make sense to me:

       "Moreover, this will not provide any meaningful insights to analysis on stability
        in that we want to measure how many changes in the perturbed explanation that
        correspond to small (and not large) alterations to the document.

  I do not follow. The top feature might change without big changes to the document.

  Suggestion: Coalesce the discussion regarding the top features into one place and present a
  self-consistent argument of where and why they are allowed to be changed or not.

Smaller things:

- The requirement of black-box access should not dismiss comparisons with white-box attacks as
  baselines.

- You write:

      "As a sanity check, we also constrain the final perturbed document to result in at least one
      of the top k features decreasing in rank."

  This does not sound like a sanity check but rather a requirement of your method. If it were
  sanity check, you'd measure whether at least one of the top k features decreased without imposing
  it as a requirement.

- The example of Table 5 seems to actually change the meaning significantly. Why was such a change
  allowed given "think" (verb) and "thinking" (most likely adjective) changed part of speech?

- You write:

      "Evidently, replacing any of the procedure steps of XAIFOOLER with a random mechanism dropped
      its performance"

  I'm unsure that "better than random" is a strong demonstration of capability.

**Reproducibility:**

3: Could reproduce the results with some difficulty. The settings of parameters are underspecified or subjectively determined; the training/evaluation data are not widely available.

**Reviewer Confidence:**

3: Pretty sure, but there's a chance I missed something. Although I have a good feel for this area in general, I did not carefully check the paper's details, e.g., the math, experimental design, or novelty.

**Typos Grammar Style And Presentation Improvements:**

- The term "ranked list" is used several times but it is not clear whether that includes scores of
  the items in the list. Please define or used a different term before first use. The description
  of Feature Weighting suggests scores are part of ranked list.

- Line 353: "explanation similarity" -> "explanation dissimilarity"

---

> ### Author Rebuttal · Authors · 2023-08-29
>
> **Summary:** The authors would like to thank the reviewer for their substantial effort in enumerating their concerns and opinions. Especially, we appreciate the extensive nature of the provided suggestions. Given the potential positive impacts of our work in promoting a more secure and robust XAI and hence encouraging the adoption of XAI in practice, we respectfully ask the reviewer to reconsider the following main concerns. First, regarding the source of instability concern. We have already varied different components in our experiments (e.g., target model, dataset) while consistently using the LIME’s default setting to minimize the confounding effects on our instability analysis. Even with these efforts, we also acknowledged in the Limitation section regarding this concern as an out-of-scope direction for future work. In addition to this concern, we recognize that the remaining provided suggestions (listed in the “reasons to reject” box) did not constitute major technical or methodology flaws of our work and hence we respectfully ask the reviewer to reconsider these in the final scoring.
>
> We appreciate and fully address the suggestive feedback from the reviewer below and also in our final manuscript.
>
>
>
> > Concerns/suggestions regarding source of instability
>
> To control model instability, we have preserved its original prediction and further explored setting a bound on the difference between output probabilities for original and perturbed texts. Initial investigations revealed only slight differences. We thus removed this constraint in the final experiments to align with general adversarial attack paradigms where attack success is based on class output rather than direct probability differences.
>
> One of our primary, if unstated goals, was to closely adhere to the “standard” LIME usage for better generalization to common practitioner use cases by avoiding changes from the default. Maintaining default LIME settings also facilitated greater-depth testing across three diverse datasets and target transformer models. While recognizing the importance of exploring alternative LIME settings and components, we consider this a natural next step for future research.
>
> We also appreciate the suggestions of directly perturbing the word embedding of the target model. While further experimentation is necessary to determine whether perturbing word embeddings would yield greater stability, our focus is to design a threat model that is realistic and practical, which often assumes that a white-box access to the target model’s parameters (e.g., embedding layers) is unavailable.
>
> Most importantly, we have *acknowledged the source of instability as one of out-of-scope topics in the Limitation section* and considered this as a future work.
>
> > Concerns regarding sample complexity of LIME’s learning model and suggestion on investigating smaller models and the unnecessity of using state-of-the-art models
>
> We use LIME’s default Logistic Regression learning model, which is a sufficiently simple model for this purpose. Most importantly, we are not trying to train LIME’s learning model to approximate the complex transformer models such as BERT and RoBERTa  as a whole but only their decision boundaries on a small, local vicinity surrounding a single data point. This requires a much smaller sampling rate. Evidently, Figure 2 shows that a much larger sampling rate is not needed as the stability reaches a plateau after the sampling rate reaches a certain threshold that is close to LIME’s default, recommended sampling rate.
>
> We here provide the following data from a simple feed-forward neural network (FNN) on two NLP datasets. Here the measure is reported in RBO (the lower the better). As we can observe, our framework is still able to perform well on a simple FNN model, achieving low RBO scores similarly to when tested with complex target models BERT and RoBERTa.
>
> | Model | Dataset | RBO |
> |---|---|---|
> | FNN | IMDB | 0.42 |
> | BERT | IMDB | 0.44 |
> | FNN | Twitter Hate Speech | 0.33 |
> | RoBERTa | Twitter Hate Speech | 0.44 |
>
> > Concerns regarding that the top features does not change and constraints in Eq. (5)
>
> The reviewer might have misunderstood our writings and we would like to clarify as follows. Our intent behind Eq. (5) is only to prevent the perturbation of features that are considered significantly important by LIME. Our primary concern is that if they are candidates for perturbation, the results become uninteresting in that changing an important feature results in an appreciable change in the explanation. We are unconcerned with the location of this feature, just that it remains within the explanation. So, we agree with the reviewer that Eq. (5) actually makes our problem more challenging and practical, hence worth investigating.
>
> > Concerns regarding dismissing comparisons with white-box attacks as baselines
>
> We intentionally scoped our paper as a black-box attack due to two reasons. First, black-box attack is more realistic in practice, which makes our analysis closer to what will happen in real life where the text perturbation process does not require parameters access to the target model. Second, black-box attack is aligned with the original purpose of LIME being a post-hoc explainer for black-box models. Regardless, our framework is also compatible with white-box attacks in NLP such as HotFlip as pointed out by the reviewer.
>
> > Concerns regarding Table 5
>
> The word replacement in table 5 is algorithmically correct, but the reviewer makes an excellent point in that the replacement might come across as not truly semantically valid without further explanation. We would like to clarify as follows. The perturbation algorithm constraints for part of speech replacement only work at word level, so while the perturbation from think -> thinking passes the algorithm’s checks as both can exist in different sentences but are the same type of word, and the resulting replacement in the sentence itself may be “out of bounds”. We will elaborate in the final manuscript to emphasize that the constraints here only apply to world level replacements.
>
> > Concerns regarding performance comparison with a random baseline
>
> In addition to the random perturbation baseline, we also carefully select other non-random baselines (Inherency, LOM, L_p)  in relevant literature as shown in our main experiment results. Moreover, we also provide an ablation study to analyze the performance gain of different components of the proposed framework.
>
> > Disjoint features requirement
>
> We want to clarify that this feature is a desired property or a requirement. Generally speaking, should a feature be selected for perturbation that occurs repeatedly within the document, the resulting perturbed explanation will have its previous length plus one for the new word now within the document.
>
> > Suggestion on using other metrics for optimization
>
> Our inclusion of multiple comparative metrics was focused towards providing a justification for RBO as an efficient measure of similarity. Our goal is chiefly to demonstrate instability in LIME, the choice of RBO is just a means to do so in a manner that satisfies the requirements we set out in Section 5.3. The other metrics have seen use in similar related work, we felt that empirical justification of RBO was necessary rather than an argument solely from desiderata of Section 5.3.
>
> > Low reproducibility score (3.0 score)
>
> We respectfully disagree with a low given reproducibility score. We have provided details on reproducibility in the Appendix section and in the main text, including specific hardwares, software, datasets, sampling rates, RBO parameters and other hyper-parameters needed to reproduce the results. We also mentioned that we will open-source all implementations. We also endeavored to maintain as close to the default settings for the packages used (e.g., LIME) as possible to avoid only a shallow investigation among different datasets and models due to concerns of the appearance of cherry picking / anomalous sampling

---

### Meta-Review · Area_Chair_EoXT · 2023-09-20

**Recommendation:** 4

**Metareview:**

This work studies the "stability" of LIME-based word importance scores. It consists in making small perturbations to the input, so as to preserve its meaning, but changing the explanation in terms of the ranking of the most important words. The perturbation technique introduced by the authors is named XAIFooler and is compared to various baseline methods, demonstrating its superiority as an adversarial attack method on LIME, and more broadly, this result highlights the "instability" or vulnerability of the LIME explanation method.

A strength of the work is that the authors use various metrics both to quantify the preservation of meaning during perturbation, as well as to quantify the resulting explanation dissimilarity. Thereby the authors make the point that among explanation similarity metrics the Rank-biased Overlap (RBO) is the most suited. Overall the evaluation is thorough, and in addition the paper is well-written.

More generally, the present work is complementary to a previous work also studying the vulnerability of LIME explanations to small perturbations in the input (Sinha et al. 2021, BlackboxNLP Workshop). Compared to this previous work, the authors additionally: 1) investigate the "inherent" instability of LIME (i.e. the impact of the LIME sampling rate on the LIME explanation similarity), 2) provide some analysis and comparisons to justify the choice of a similarity metric for explanations, and finally 3) allow only perturbation of less important words to demonstrate the instability of LIME even when performing only subtle changes in the input.

Nevertheless it remains an open question for future work whether the discovered explanation instabilities are specific to the LIME explanation method, or if such instabilities arise from the LIME's explanation setup which could also apply to other explanation methods as well, i.e. having only black-box access to the model and performing only a limited number of queries onto the model (through the sampling rate).

---

### Decision · Program_Chairs · 2023-10-07

**Decision:**

Accept-Main

**Comment:**

This work studies the "stability" of LIME-based word importance scores. It consists in making small perturbations to the input, so as to preserve its meaning, but changing the explanation in terms of the ranking of the most important words. The perturbation technique introduced by the authors is named XAIFooler and is compared to various baseline methods, demonstrating its superiority as an adversarial attack method on LIME, and more broadly, this result highlights the "instability" or vulnerability of the LIME explanation method.

A strength of the work is that the authors use various metrics both to quantify the preservation of meaning during perturbation, as well as to quantify the resulting explanation dissimilarity. Thereby the authors make the point that among explanation similarity metrics the Rank-biased Overlap (RBO) is the most suited. Overall the evaluation is thorough, and in addition the paper is well-written.

More generally, the present work is complementary to a previous work also studying the vulnerability of LIME explanations to small perturbations in the input (Sinha et al. 2021, BlackboxNLP Workshop). Compared to this previous work, the authors additionally: 1) investigate the "inherent" instability of LIME (i.e. the impact of the LIME sampling rate on the LIME explanation similarity), 2) provide some analysis and comparisons to justify the choice of a similarity metric for explanations, and finally 3) allow only perturbation of less important words to demonstrate the instability of LIME even when performing only subtle changes in the input.

Nevertheless it remains an open question for future work whether the discovered explanation instabilities are specific to the LIME explanation method, or if such instabilities arise from the LIME's explanation setup which could also apply to other explanation methods as well, i.e. having only black-box access to the model and performing only a limited number of queries onto the model (through the sampling rate).